# Three-Dimensional Paper-Based Microfluidic Analysis Device for Simultaneous Detection of Multiple Biomarkers with a Smartphone

**DOI:** 10.3390/bios10110187

**Published:** 2020-11-21

**Authors:** Seung Ho Baek, Chanyong Park, Jaehyung Jeon, Sungsu Park

**Affiliations:** 1School of Mechanical Engineering, Sungkyunkwan University, Suwon 16419, Korea; steeve777777@gmail.com (S.H.B.); jjh4760.official@gmail.com (J.J.); 2Department of Biomedical Engineering, Sungkyunkwan University, Suwon 16419, Korea; cksdyd6348@naver.com; 3Biomedical Institute for Convergence at SKKU (BICS), Sungkyunkwan University, Suwon 16419, Korea

**Keywords:** paper-based microfluidic device, colorimetric, multiple detection, smartphone application

## Abstract

Paper-based microfluidic analysis devices (μPADs) have attracted attention as a cost-effective platform for point-of-care testing (POCT), food safety, and environmental monitoring. Recently, three-dimensional (3D)-μPADs have been developed to improve the performance of μPADs. For accurate diagnosis of diseases, however, 3D-μPADs need to be developed to simultaneously detect multiple biomarkers. Here, we report a 3D-μPADs platform for the detection of multiple biomarkers that can be analyzed and diagnosed with a smartphone. The 3D-μPADs were fabricated using a 3D digital light processing printer and consisted of a sample reservoir (300 µL) connected to 24 detection zones (of 4 mm in diameter) through eight microchannels (of 2 mm in width). With the smartphone application, eight different biomarkers related to various diseases were detectable in concentrations ranging from normal to abnormal conditions: glucose (0–20 mmol/L), cholesterol (0–10 mmol/L), albumin (0–7 g/dL), alkaline phosphatase (0–800 U/L), creatinine (0–500 µmol/L), aspartate aminotransferase (0–800 U/L), alanine aminotransferase (0–1000 U/L), and urea nitrogen (0–7.2 mmol/L). These results suggest that 3D-µPADs can be used as a POCT platform for simultaneous detection of multiple biomarkers.

## 1. Introduction 

Paper-based microfluidic analysis devices (μPADs) have attracted attention as a cost-effective tool for point-of-care testing (POCT), food safety, and environmental monitoring [1,2,3,4]. They are inexpensive, easy to use, and do not require any external power source. Recently, three-dimensional (3D)-μPADs have been developed to improve the performance of two-dimensional (2D)-μPADs [5,6]. In 3D-µPADs, the combination of lateral and vertical flows shortens analysis times and enables multiple steps, allowing for the integration of various functions [7]. In addition, unlike 2D-µPADs, 3D-µPADs have the advantage of fewer sample loss since the top of the channel is covered, minimizing evaporation of the sample while the fluid flows through the channel [8]. However, conventional fabrication methods for 3D-μPADs are labor-intensive and cumbersome because they require the use of additional materials on the exterior, including double-sided sticky tape and adhesive spray [9] to bond two pieces of paper.

Most recently, we have reported on a method to fabricate 3D-µPADs for the colorimetric detection of biomarkers using 3D digital light processing (DLP) printing [10]. By exposing photocurable resin to be cross-linked with visible light, 3D hydrophobic structures inside and outside the cellulose paper can be printed. They were printed within 2 min without any extra bonding steps and did not show leakage of the solution, which was observed in other 3D-µPADs [11,12]. POCT kits can be used to accurately diagnose a specific disease by comprehensively analyzing the detection results of several biomarkers [13]. For example, bone-related diseases may be suspected when very high levels of alkaline phosphatase (ALP) are detected [14] whereas fatty liver or viral hepatitis may be suspected when very high levels of both aspartate aminotransferase (AST) and ALP are detected [15]. To demonstrate the feasibility of the 3D-µPADs as a POCT platform, it has to be able to detect more biomarkers at the same time, and the detection results must be more easily and quickly quantified.

Smartphones can provide easy detection and quick analysis results for POCT kits [16,17,18]. For example, smartphone-based colorimetric and fluorescence analyses have been extensively studied for medical diagnostics [19,20], environmental monitoring [21], and food evaluation [22]. So far, only a small number of biomarkers can be analyzed using a smartphone, making it difficult to accurately diagnose diseases. Therefore, smartphones should be able to analyze multiple samples for accurate diagnoses of diseases [23,24] in the future.

In this study, we report on 3D-µPADs for smartphone-based detection of multiple biomarkers. The 3D-µPADs were fabricated using a 3D DLP printer with ultraviolet (UV) light. They consisted of a single reservoir connected with 24 detection zones through eight microchannels, allowing eight different biomarkers to be simultaneously and repeatedly detected in a sample. Colored signals from the 3D-µPADs were imaged and analyzed using a smartphone.

## 2. Materials and Methods

### 2.1. Chemicals

D-(+)-Glucose, glucose oxidase (GOx) from Aspergillus niger, 4-aminoantipyrine (4-AAP), albumin from human serum, tetrabromophenol blue (TBPB), citric acid, cholesterol (Chol.) ALP, alkaline phosphatase yellow (p-nitrophenyl phosphate), an AST activity assay kit, an alanine aminotransferase (ALT) activity assay kit, a creatinine assay kit, and ethyl alcohol were purchased from Sigma-Aldrich Korea (Seoul, Korea). A urea nitrogen (UN) colorimetric detection kit was purchased from Thermo Fisher Scientific (San Jose, CA, USA). Horseradish peroxidase (HRP), cholesterol oxidase (CO), and cholesterol esterase (CE) were obtained from Toyobo Co. (Osaka, Japan). N-Ethyl-N-(2-hydroxy-3-sulfopropyl)-3,5-dimethylaniline sodium salt monohydrate (MAOS), N-ethyl-N-(2-hydroxy-3-sulfopropyl)-3-methoxyaniline, sodium salt, and dihydrate (ADOS) were purchased from Dojindo (Rockville, MD, USA). 

### 2.2. Fabrication of 3D-µPADs

The top and bottom structures of the 3D-µPADs were separately designed using the Student Edition of Inventor^®^ Professional (Autodesk Inc., San Rafael, CA, USA) and saved as different STL files. The files were then sent to the DLP printer (IM1™, Carima Co., Seoul, Korea) using a universal serial bus (USB) drive.

Before printing, a layer of filter paper (Qualitative filter paper: grade 1) from Whatman International Ltd. (Maidstone, UK) was soaked with UV-curable resin (CFY044W, Carima Co.) in a petri dish for 10 s. The resin-soaked paper was moved to a tray filled with resin in the printer. Then, its bottom side was exposed for 2 s to the UV patterns for the bottom structures projected through the tray (Figure 1b). After the bottom printing, the paper was turned upside down at the same spot of the plate that had been marked and its bottom side was exposed for 60 s to the ultraviolet (UV) patterns for the upper structures projected through the tray (Figure 1). Next, the printed paper was removed from the tray and then rinsed by 100% ethyl alcohol for 20 min with gently shaking. Finally, the paper was air-dried for 1 min at room temperature.

### 2.3. Enzyme Fixation

The reagents for the detection of glucose were prepared by dissolving MAOS (1 mM), 4-AAP (10 mM), HRP (1 mg/mL), and GOx (10 mg/mL) in phosphate-buffered saline (PBS: pH 7.4) [23,25]. The reagent for the detection of albumin was prepared by dissolving TBPB (3.3 mM) and citric acid (0.1 mM) in 95% ethyl alcohol [26]. The reagent used to detect Chol. was prepared by dissolving ADOS (1 mM), CE (14.1 U), CO (10 mg/mL), 4-AAP (10 mM), and HRP (1 mg/mL) in PBS [10]. The reagent used to detect ALP was prepared by dissolving para-nitrophenylphosphate (1.35 mM) in PBS [27]. The remaining AST, ALT, creatinine, and UN enzymes were prepared according to commercial assay kit manuals. Finally, 2 µL of each enzyme solution was dropped into its designated detection zone using a micropipette and air-dried for its immobilization at room temperature (RT). 

### 2.4. Mechanism of Colorimetric Assay

To test whether the 3D-µPADs were able to simultaneously detect multiple biomarkers, eight biomarkers for metabolic diseases such as diabetes and cardiovascular, liver, and renal diseases were selected. Glucose is a biomarker for diabetes [28], cholesterol for cardiovascular disease [29], albumin, ALT, ALP, and AST for liver diseases [30], albumin, UN, and creatinine for renal diseases [31]. Colorimetric reaction through enzymes is the core mechanism of multiple biomarkers. For colorimetric detection of glucose, the GOx precipitated in the detection zone, which oxidizes glucose to gluconic acid while generating hydrogen peroxide. Another enzyme occupied in the detection zone, known as HRP, converted hydrogen peroxide into oxygen. Simultaneously, 4-AAP was reacted with MAOS. At the end of the HRP-mediated reaction, a blue product was formed [23]. For colorimetric detection of albumin, the TBPB precipitated in the detection zone, which reacts like a pH indicator. The higher the amount of albumin, the more it reacts with TBPB, causing a color change from yellow to blue as the pH increases [32]. For colorimetric detection of cholesterol, the CO and CE precipitated in the detection zone, which oxidizes cholesterol to Chol-4-en-3-one while generating hydrogen peroxide. Another enzyme occupied in the detection zone, HRP, converted hydrogen peroxide into oxygen. Simultaneously, 4-AAP reacted with ADOS. At the end of the HRP-mediated reaction, a pink product was formed. For colorimetric detection of ALP, the para-nitrophenylphosphate precipitated in the detection zone. As albumin and para-nitrophenylphosphate react, a yellow product, known as p-nitrophenylate, was formed. The remaining AST, ALT, creatinine, and UN reaction follows the commercial assay kit mechanism.

### 2.5. Detection of Biomarkers in PBS 

To obtain a dose-response curve for biomarkers, standard solutions of the biomarkers were prepared by mixing the stock solution of each biomarker with different volumes of PBS (pH 7.4). The stock solution was prepared by dissolving a certain concentration of a biomarker in PBS. The concentrations of the standard solutions for the biomarkers varied, depending on the optimal detection range of each biomarker: 0–20 mmol/L for glucose, 0–10 mmol/L for Chol., 0–7 g/dL for albumin, 0–800 U/L for ALP, 0–500 µmol/L for creatinine, 0–800 U/L for AST, 0–1000 U/L for ALT, and 0–7.2 mmol/L for UN. To generate the standard curve of a biomarker, 10 µL of the standard solution of the biomarker was dropped on the detection zone. At 1 min after the dropping, the gray intensity in the detection zone was measured using ImageJ (NIH, Bethesda, MD, USA). 

To simultaneously detect the biomarkers in PBS, a mixture of the biomarkers at different concentrations was prepared by mixing their stock solutions with PBS at various ratios. Furthermore, 300 µL of the mixture was dropped onto the sample reservoir of the 3D-µPADs. At 3 min after the dropping, the gray intensity of each detection zone was measured using ImageJ.

### 2.6. Operation of the Smartphone Application 

Once the names of the biomarkers were marked on the start screen, the linearization equation calculated from the dose-response curve was specified with its normal range. Images of the control and test samples were imported into the application. The detection zones in the images were designated as circles. The gray intensity in each circle was measured. Then, the gray intensity value for each biomarker was obtained by subtracting the gray intensity of the sample from the intensity of the control. Eventually, the concentration and level (low to high) of the eight biomarkers appeared.

### 2.7. Photo Box for Capturing Images from 3D-µPADs

To reduce the effect of reflecting light on the measurement of the color intensity of the detection zones [33], a switchable white light bulb was installed in a black cardboard box (measuring 90 mm by 60 mm by 40 mm in width, length, and height, respectively) with a sliding door, allowing the 3D-µPADs to be inserted into the box.

## 3. Results and Discussion

### 3.1. Effect of Exposure Time on the Height of Cured Resin

For fast printing, UV light instead of visible light [8] was used to cure the resin to fabricate 3D-µPADs. Since the hydrophilic surface of paper is changed to a hydrophobic one with curing by UV light, the degree of curing can be estimated by changing the color of the cross section of the paper with red ink when the ink was dropped onto the paper. As shown in Figure 2a, the cross section of the uncured paper was completely wet with red ink. At 1 s, ~50 µm of the paper was wet with the ink, indicating that part of it was cross-linked. This uniformity was further supported by the scanning electron microscope images showing fibrous structures in uncured parts, which disappeared in the cured parts (Appendix A). At 2 s, ~90 µm of the paper was cured. At 3 s, the paper was not wet with the ink, exhibiting only the white color of the cross-linked resin. The thickness of the cured paper was ~200 µm, which is greater than the thickness of the original paper (180 µm), indicating that the paper was slightly overexposed. Based on the results, the optimal exposure time for fabricating the bottom structures inside the paper was chosen to be 2 s, which half of the paper was cured. The height of the structure cured by the resin in this study increased with rising exposure time. This is because the amount of cross-linked resin increases as the UV exposure time increases [10,20]. In fact, the reservoir at 5 mm height was printed by overexposing the paper for 60 s (Figure 2b). It takes about 90 s to print the top and bottom structures of the paper and remove uncured resin from the paper and the printing costs less than a dollar per device when the devices are mass produced. The results suggest that our new method is highly suitable for the rapid production of 3D-µPADs.

The 3D-µPADs consist of a sample reservoir (diameter: 12 mm, height: 5 mm) in the center connected with 24 detection zones (diameter: 4 mm) through eight channels (width: 2 mm, length: 6 mm) (Figure 2c). At least 300 µL of solution needed to be loaded into the sample reservoir to fully wet all the detection zones. By printing the reservoir 5 mm high, a large amount of sample solution up to 500 µL can be loaded into the device without overflow, which was often observed in 2D-µPADs [34]. With this increase, it is suggested that the 3D-µPADs are highly suitable for handling a sample in a large volume, thereby enabling analysis of multiple biomarkers.

### 3.2. Uniform Distribution of Fluid on 3D-µPADs

The flow rate of a sample solution affects signal generation in 3D-µPADs [35]. Therefore, the channel size in the 3D-µPADs should be uniform to reduce signal variation in the 3D-µPADs. The quality of the printed structures in the 3D-µPADs was examined by measuring the height of the channels in the 3D-µPADs (Figure 2d). The average channel height was 90 ± 1.2 μm (*n* = 48), indicating that all channels of the 3D-µPADs were uniformly fabricated.

To test whether solutions in the reservoir reached the detection zones at a similar flow rate, the flow speed of ink in each channel was measured after dropping 300 µL of red ink into the reservoir over the channels in the 3D-µPADs at different times (0–150 s) (Figure 2e). It took 150 s for the ink to travel 20 mm, the distance between the detection zone and the center of the reservoir, corresponding to a speed of 0.13 mm/s. The ink in all 24 channels traveled through the paper at a similar speed. Taken together, our results suggest that our printing method can rapidly fabricate 3D-µPADs with high accuracy.

### 3.3. Quantitative Detection of Analytes in PBS

A dose-response curve for each biomarker was obtained by quantifying the gray intensity in the detection zones at different concentrations of biomarkers, as shown in Figure 3. Five experiments were repeated to draw a dose-response curve, and the error of this result was expressed as a standard deviation. The gray intensity was proportional to the biomarker concentration. The shape of the dose-response curve of all biomarkers follows a quadratic parabolic equation with a saturation point, and the reliability (R^2^) values were all >0.96. According to the dose-response curves, the limits of detection were 0.3 mmol/L for glucose, 0.3 mmol/L for cholesterol, 0.1 g/dL for albumin, 10 U/L for ALP, 40 μmol/L for creatinine, 10 U/L for AST, 50 U/L for ALT, and 0.04 mmol/L for UN. The linear range of each biomarker was obtained from the dose–response curve, including its respective normal range (2.5–5.8 mmol/L for glucose [25,36], 5.2–6.2 mmol/L for cholesterol [37], 3.4–5.4 g/dL for albumin [38], 44–147 U/L for ALP [39], 45–90 μmol/L for creatinine [40], 0–40 U/L for AST [41], 7–56 U/L for ALT [42], and 2.5–7 mmol/L for UN [43]) Coefficient of variance (CV) values at each concentration of all biomarkers were less than 10%, showing good reproducibility. These results suggest that the 3D-µPADs are quite useful for quantification of all eight biomarkers as well as for the diagnosis of metabolic diseases. 

### 3.4. Uniform Signal at Each Detection Zone on 3D-µPADs

The 3D-µPADs have 24 detection zones in total, with three detection zones in a pair allocated to each biomarker, allowing the biomarker to be measured three times (Figure 4a). When a sample solution in PBS containing all eight biomarkers was loaded into the sample reservoir, similar color intensities were observed from the detection zones for each biomarker (Figure 4b–i). Analysis of gray intensities in the zones showed that there were low variations in the three detection zones, which was supported by coefficients of variation of <5% (being 2.7% for glucose, 3% for cholesterol, 4% for albumin, 4.3% for ALP, 4.8% for creatinine, 4.6% for AST, 1.2% for ALT, and 1.1% for UN). These low variations in the detection zone could be due to the relatively higher accuracy in printing channel structures (Figure 2d) accompanied by a uniform flow speed (Figure 2e) compared to conventional µPADs [44,45,46].

### 3.5. Simultaneous Detection of Multiple Biomarkers in PBS Using the Smartphone Application

For fast and convenient detection of multiple biomarkers on the 3D-µPADs, a smartphone application was developed and tested with 300 µL of PBS containing the eight biomarkers (8 mmol/L for glucose, 8 mmol/L for cholesterol, 4 g/dL for albumin, 400 U/L for ALP, 150 μmol/L for creatinine, 400 U/L for AST, 400 U/L for ALT, and 4 mmol/L for UN). Once the colors were generated from the detection zones on the 3D-µPADs, which took ~3 min (Figure 5a,b), the image of the 3D-µPADs was captured in the photo box using the smartphone (Figure 5c–e). Then, the gray intensities in the detection zones were measured and the concentrations of the biomarkers were calculated by using the application (Figure 5f–h). The concentrations obtained by the 3D-µPADs with the smartphone application were not significantly different from the original concentrations in PBS, which was indicated by a relative error of <5% (Table 1). These concentrations were similar to those obtained by ImageJ, exhibiting a relative error of <7% (Table 1). Depending on the concentration, the level of the biomarkers in the sample was determined to be low, normal, or high (Figure 5h). Similarly, the application could be used for the simultaneous detection of multiple biomarkers in serum samples (Appendix A). This was demonstrated by small differences between the gray intensities of the detection zones and a serum sample obtained by ImageJ and the application (Appendix A). These results suggest that the 3D-μPADs with the application are highly useful for multiple detections of biomarkers and convenient self-diagnosis of diseases.

## 4. Conclusions

In this study, 3D-µPADs were fabricated using a 3D DLP printer for multiple detections. The reservoir on 3D-µPADs can store enough sample for multiple detections. Sequentially, the concentration can be obtained three times per biomarker in one experiment to reduce errors. Eight biomarkers (glucose, cholesterol, albumin, ALP, creatinine, AST, ALT, and UN) in PBS and serum could be detected simultaneously on the 3D-μPADs through a smartphone. In this experiment, the size of the detection zone was set to 4 mm to easily analyze the selected area using a smartphone camera. However, using a high-quality smartphone, we could minimize the detection zone. This also reduces the amount of the sample, which leads to the detection of more biomarkers. In previous research, we succeeded in integrating plasma separation membrane with 3D-µPADs [20]. This suggests that the 3D-µPADs with the plasma separation membrane will allow patients to easily diagnose diseases remotely using a smartphone at home in the future.

## Figures and Tables

**Figure 1 biosensors-10-00187-f001:**
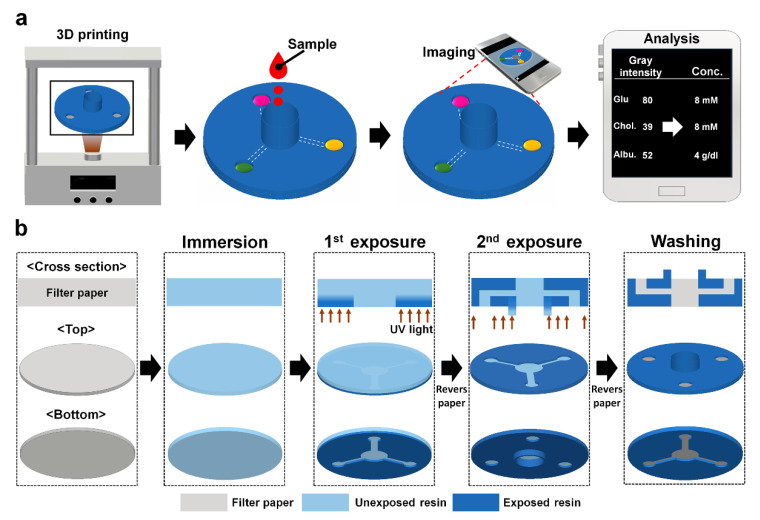
Schematics describing (**a**) smartphone-based colorimetric detection for multiple biomarkers on the 3D-μPADs and (**b**) 3D DLP printer-based fabrication of the 3D-μPADs.

**Figure 2 biosensors-10-00187-f002:**
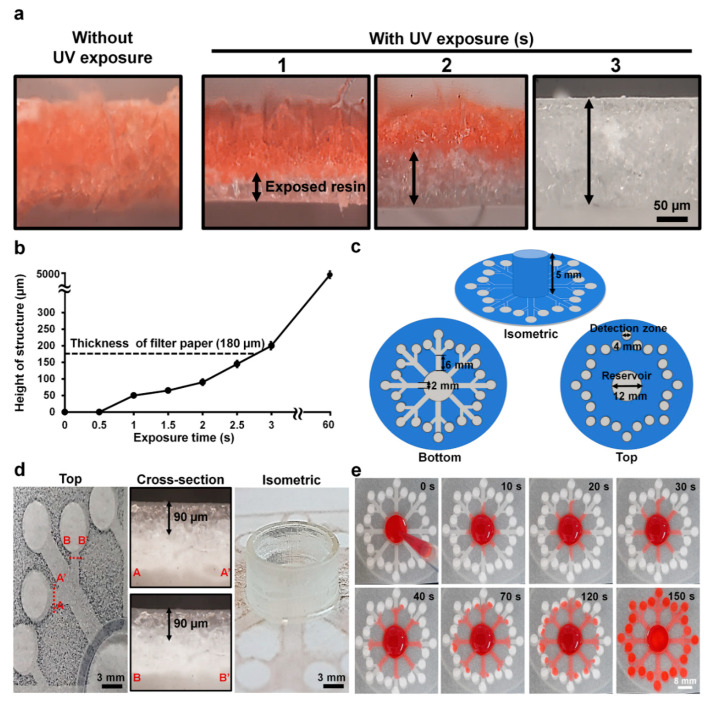
Control of cross-linked structures and flow inside filter paper. (**a**) Cross-sectional views of the resin inside the paper with and without ultraviolet (UV) exposure at different times (1–3 s). The black arrow indicates exposed resin areas. (**b**) Height of cross-linked resin structures inside the paper at different exposure times (0–60 s). The dotted black line indicates the thickness of the paper. (**c**) Bottom, top, and isometric views with dimensions of the design for the 3D-μPADs. (**d**) Cross-sectional images of two different channels and an isometric image of a sample reservoir. The black arrow indicates exposed resin that is the height of the structure. (**e**) Flow distribution in the 3D-μPADs at different times (0–150 s). For visualization of flow, red ink (300 μL) was added into the sample reservoir (center).

**Figure 3 biosensors-10-00187-f003:**
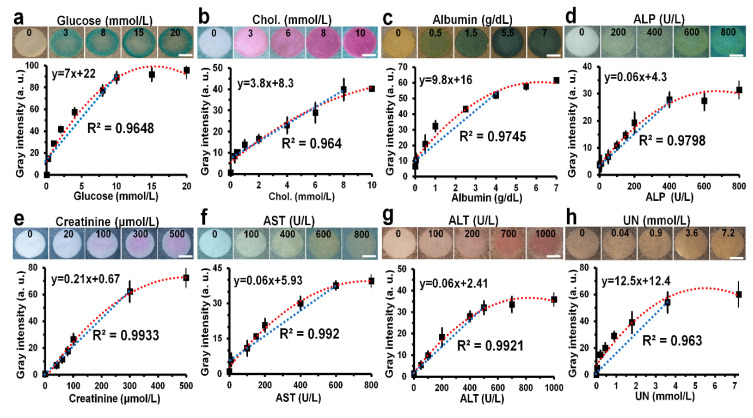
Colorimetric detection of eight biomarkers (glucose, Chol., albumin, ALP, creatinine, AST, ALT, and UN) in PBS. (**a**–**h**) Dose-response curves were calculated from the data obtained by 15 repeated uses of each analyte at different concentrations and representative images of the detection zones reacted with various concentrations of a single biomarker: glucose (0–20 mmol/L), Chol. (0–10 mmol/L), albumin (0–7 g/dL), ALP (0–800 U/L), creatinine (0–500 μmol/L), AST (0–800 U/L), ALT (0–1000 U/L), and UN (0–7 mmol/L). (Scale bar: 4 mm. a.u.: arbitrary units.).

**Figure 4 biosensors-10-00187-f004:**
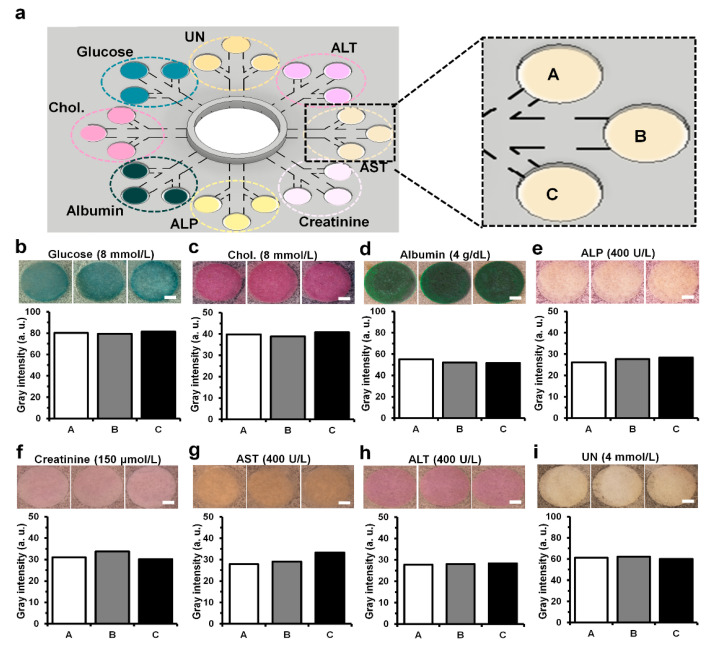
Uniformity comparison of colorimetric detection of eight biomarkers (glucose, cholesterol, albumin, ALP, creatinine, AST, ALT, and UN) at the concentration in PBS. (**a**) Schematics of the 3D-μPADs for simultaneous detection of the eight biomarkers. The 3D-μPADs have three detection zones (A, B, and C) per each biomarker. (**b**–**i**) Gray intensities calculated from the data obtained by three repeated uses of each analyte at the concentrations and representative images of the detection zones reacted with the concentrations of a single biomarker: glucose (8 mmol/L), cholesterol (8 mmol/L), albumin (4 g/dL), ALP (400 U/L), creatinine (150 μmol/L), AST (400 U/L), ALT (400 U/L), and UN (4 mmol/L). (Scale bar: 1 mm. a.u.: arbitrary units. *n* = 3).

**Figure 5 biosensors-10-00187-f005:**
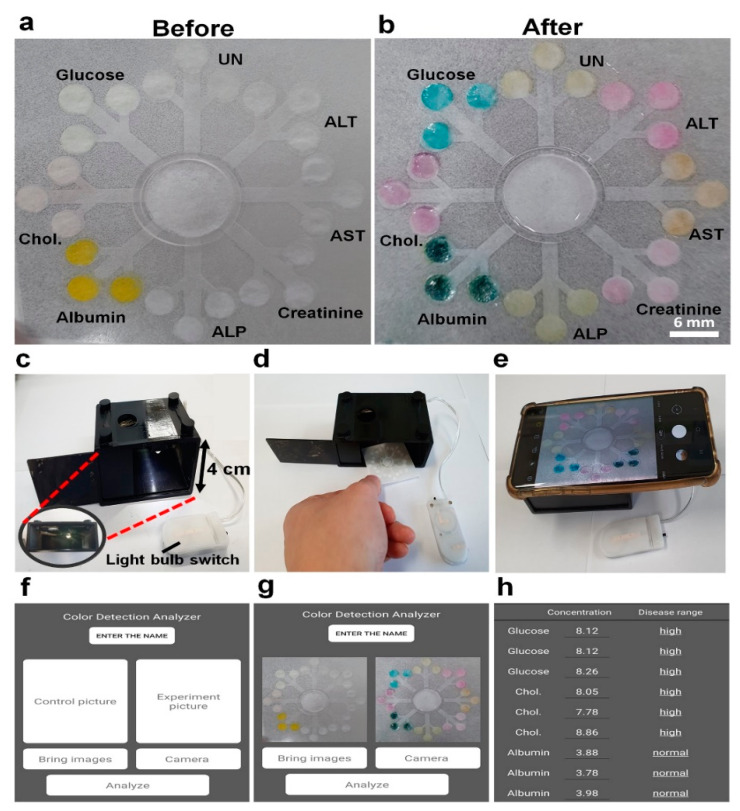
Schematics for detection of multiple biomarkers on the 3D-μPADs in the photo box using the smartphone application. (**a**) Each enzyme is fixed on each detection zone before loading the sample. (**b**) Simultaneous detection of multiple biomarkers including 8 mmol/L for glucose, 8 mmol/L for Chol., 4 g/dL for albumin, 400 U/L for ALP, 150 μmol/L for creatinine, 400 U/L for AST, 400 U/L for ALT, and 4 mmol/L for UN in PBS. (**c**–**h**) Using the smartphone application to measure the concentration and normality of each biomarker.

**Table 1 biosensors-10-00187-t001:** Concentrations of biomarkers based on gray intensities measured by ImageJ and the smartphone application with the photo box. Biomarkers at different concentrations were prepared by dissolving known concentrations of the biomarkers in PBS before testing. The mean and standard deviation of each biomarker were obtained from the measurement of gray intensity on three different µPADs devices with nine detection zones. Device-to-device relative standard deviation (RSD) was less than 5% (1.2% for glucose, 3.2% for cholesterol, 4.7% for albumin, 2.8% for ALP, 3.7% for creatinine, 0.9% for AST, 4.4% for ALT, and 3.1% for UN).

Biomarker Concentration [A]	ImageJ		Smartphone Application	
	Concentration [B] ^⊥^	Relative Error (%) [(∣A − B∣/A) × 100]	Concentration [C] ^⊥^	Relative Error (%) [(∣A − C∣/A) × 100]
**High glucose** **(8 mmole/L)**	8.3 ± 0.1 (mmole/L)	3.8	8.2 ± 0.2 (mmole/L)	2.1
**High Cholesterol (8 mmole/L)**	8.5 ± 0.3 (mmole/L)	6.3	8.2 ± 0.5 (mmole/L)	2.9
**Normal albumin** **(4 g/dL)**	3.8 ± 0.2 (g/dL)	5.7	3.9 ± 0.11 (g/dL)	3.1
**High ALP (400 U/L)**	386.6 ± 20.2 (U/L)	3.4	402.2 ± 18 (U/L)	0.6
**High creatinine (150 µmole/L)**	147.3 ± 8.9 (µmole/L)	1.8	150.6 ± 7.4 (µmole/L)	0.5
**High AST** **(400 U/L)**	402.6 ± 47.8(U/L)	0.6	395.8 ± 34.7 (U/L)	1.1
**High ALT (400 U/L)**	426.8 ± 5.7 (U/L)	6.7	415.7 ± 9.6 (U/L)	3.9
**Normal UN** **(4 mmole/L)**	3.9 ± 0.1 (mmole/L)	2.5	4 ± 0.1 (mmole/L)	0.1

**^⊥^** The concentration was calculated from the standard curve of each biomarker, as shown in Figure 3.

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
