# Peer review of "Three-Dimensional Paper-Based Microfluidic Analysis Device for Simultaneous Detection of Multiple Biomarkers with a Smartphone"

_biosensors, 2020, doi:10.3390/bios10110187_

Round 1

Reviewer 1 Report

The authors present a uPAD for biomarker detection using a smartphone-approach. Overall a good study but there are some 

1) Figure 4: add information on n=?, technical repeats only?

2) Figure 5: add experimental repeats to show variability (n>3), device-to-device and batch to batch %RSD?

3) The authors should provide analysis of whole blood at least for spiked samples to show that the current uPAD performs as demonstrated and outlined as POC device. Does it work with finger tip volumes 20 ul too -if not: how do the authors envision that a patient uses these uPADs? Dont forget to add positive control spots/lines for 100% achievable signal and QC

Reviewer 2 Report

The paper reports an interesting study on 3D printed Microfluidic, which integrates a paper for rapid test like lateral flow concepr. The described method paved the way to further application in biomedical fields and technologies. Also if it is well written and contains significant improvements and results for the research in the field, there are major improvements to be done prior to the publication /re-submissions. the Authors should revise the manuscript in order to clarify different point of strength of their method, which are:

  • At figure 1 it seems that the second exposure is operated with a double side tool, but how is possible without a double exposure machine? Please re-edit the picture properly.
  • At pag 3 line 88 “…after the bottom printing, the paper was turned upside down and its bottom side was exposed to the UV patterns for the upper structures projected through the tray.” How do you align the top and bottom structures?
  • The 3D images in figure 2 seem to be at poor resolution, also the quotations have high contrast
  • The meaning of uncured paper is not clear and should be better specified. Is it the resin that is cured / uncured and not the paper, right? Page 5 line 147

Minor:

  • Authors declare the cost effective solution for additive manufacturing, but does not include an analysis on the cost and time related to the applied surface modification technics.
  • Please be careful to Typos errors and missing space.
  • For completeness I think the authors should cite other works which have faced the issue of substrate adhesion between printed objects and polymeric structures for bio-applications in the field of additive manufacturing. For example: Title: Numerical and experimental evaluation of SLA polymers adhesion for innovative bio-MEMS, Source: MATERIALS TODAY-PROCEEDINGS  Volume: 7  Pages: 572- 577  DOI: 10.1016/j.matpr.2018.12.010  Part: 1  Published:2019

Author Response

Reviewer #2

Comment 1.

At figure 1 it seems that the second exposure is operated with a double side tool, but how is possible without a double exposure machine? Please re-edit the picture properly.

Response

We re-edit Figure 1 by adding the reverse process as below.

Comment 2.

At pag 3 line 88 “…after the bottom printing, the paper was turned upside down and its bottom side was exposed to the UV patterns for the upper structures projected through the tray.” How do you align the top and bottom structures?

Response

Using the plate of the DLP printer, we marked at the plate and the paper to algin at the same place. Even if we turn the paper upside down, we know the same place by markings. As you suggested it, we provide detail information in manuscript (from page 2 line 85~86)

Before the revision:

“After the bottom printing, the paper was turned upside down at the and its bottom side was to the UV patterns for the upper structures projected through the tray”

After the revision:

“After the bottom printing, the paper was turned upside down at the same spot of the plate that had been marked and its bottom side was exposed for 60 s to the UV patterns for the upper structures projected through the tray “

Comment 3.

The 3D images in figure 2 seem to be at poor resolution, also the quotations have high contrast/

Response

As the reviewer suggestion, I higher the resolution and lower the contrast of the figure 2a.

Comment 4.

The meaning of uncured paper is not clear and should be better specified. Is it the resin that is cured / uncured and not the paper, right?

Response

As the reviewer mentions, we edit it more specifically. (from page 4 line 151)

Before the revision:

“Because the hydrophilic surface of uncured paper is changed to a hydrophobic one with curing by UV light, the degree of curing can be estimated by changing the color of the cross section of the paper with red ink when the ink was dropped onto the paper”

After the revision:

“Because the hydrophilic surface of paper is changed to a hydrophobic one with curing by UV light, the degree of curing can be estimated by changing the color of the cross section of the paper with red ink when the ink was dropped onto the paper”

Comment 5. (Minor)

Authors declare the cost effective solution for additive manufacturing, but does not include an analysis on the cost and time related to the applied surface modification technics

Response

In the manuscript we include the time and also the cost effective of our 3D-µPAD devise as below.

“It takes about 90 s to print the top and bottom structures of the paper and remove uncured resin from the paper (Figure 2b) and the printing costs less than a dollar per device when the devices are mass produced.” (from page 4 line 165~167)

Comment 6. (Minor)

Please be careful to Typos errors and missing space.

Response

Thank you. We tried to correct them.

Comment 7. (Minor)

For completeness I think the authors should cite other works which have faced the issue of substrate adhesion between printed objects and polymeric structures for bio-applications in the field of additive manufacturing. For example: Title: Numerical and experimental evaluation of SLA polymers adhesion for innovative bio-MEMS, Source: MATERIALS TODAY-PROCEEDINGS Volume: 7 Pages: 572-

Response

As the reviewer suggest, I added the citation of the following paper.

Reviewer 3 Report

I suggest it to be accepted after minor revision. In Part 3.4 of this manuscript, the authors didn't investigate the effect of errors on the experiment quality so these data need to be added.

Apart from that, Line 88 and 89: exposure time was not mentioned. Line 90: rising method is not described. Line 105-107: Al is for 2 different types of diseases. Figure 2: the result is confusing - a and b doesn't match, and b and e results are different.

Author Response

Comment 1.

Part 3.4 of this manuscript, the authors didn't investigate the effect of errors on the experiment quality so these data need to be added.

Response

As the reviewer suggest the investigate the effect of errors, we added more information at Part 3.3. The part 3.4 are connected at part 3.3 so we add more information at part 3.3.

After the revision:

“Five experiments were repeated to draw a dose-response curve, and the error of this result was expressed as a standard deviation.” (from page 6 line 200~202)

“Coefficient of variance (CV) values at each concentration of all biomarkers were less than 10%, showing good reproducibility.” (from page 6 line 210~211)

Comment 2.

Apart from that, Line 88 and 89: exposure time was not mentioned.

Response

As the reviewer suggest, I added the exposure time. (from page 2 line 86)

Before the revision:

“After the bottom printing, the paper was turned upside down at the and its bottom side was to the UV patterns for the upper structures projected through the tray”

After the revision:

“After the bottom printing, the paper was turned upside down at the same spot of the plate that was marked and its bottom side was exposed for 60 s to the UV patterns for the upper structures projected through the tray “

Comment 3.  

Line 90: rising method is not described.

Response

As the reviewer suggest, I added the rinsing method. (from page 2 line 88)

Before the revision:

“Next, the printed paper was removed from the tray and rinsed ten times with 100% ethyl alcohol”

After the revision:

“Next, the printed paper was removed from the tray and then rinsed by 100% ethyl alcohol for 20 min with gently shaking”

Comment 4.

Line 105-107: Al is for 2 different types of diseases.

Response

Is Al means albumin? We have already indicated that albumin has 2 different types of diseases (from page 3 line 106~107)

“albumin, ALT, ALP, and AST for liver diseases; albumin, UN, and creatinine for renal diseases”

Comment 5.

Figure 2: the result is confusing - a and b doesn't match, and b and e results are different.

Response

Reviewer 4 Report

The manuscript “Three-dimensional paper-based microfluidic analysis device for simultaneous detection of multiple biomarkers with a smartphone” by Seung Ho Baek, Chanyong Park, Jaehyung Jeon and Sungsu Parkis is dedicated to simultaneous detection of different biomarkers with the use of smartphone as an optical signal processing element.

The article is good written and valuable however I have one doubt:

as the presented manuscript concerns the detection of selected, different, biomarkers (glucose, cholesterol, albumin, alkaline phosphatase, creatinine, aspartate aminotransferase, alanine aminotransferase, urea nitrogen) which differ in its nature, for me, information omitted by Authors about the detection mechanisms, reactions, responsible for color change, is serious oversight. In my opinion this information, togheter with appropriate schemes, should be added to the manuscript. Especially as the manuscript is considered for publication in “biosensors” journal where sometimes only for one such reaction the whole article is dedicated where the reaction optimization is conducted (both with possible interferants, conditions for reaction etc.).

After all in my opinion the article is interesting and after minor changes (providing some more comprehensive explanations of the developed assay and its advantages over other similar devices) it should be considered for publication in Biosensors (IF=3.2).

Author Response

omment 1.

as the presented manuscript concerns the detection of selected, different, biomarkers (glucose, cholesterol, albumin, alkaline phosphatase, creatinine, aspartate aminotransferase, alanine aminotransferase, urea nitrogen) which differ in its nature, for me, information omitted by Authors about the detection mechanisms, reactions, responsible for color change, is serious oversight. In my opinion this information, togheter with appropriate schemes, should be added to the manuscript. Especially as the manuscript is considered for publication in “biosensors” journal where sometimes only for one such reaction the whole article is dedicated where the reaction optimization is conducted (both with possible interferants, conditions for reaction etc.).

Response

As the reviewer suggest, I added the mechanism of colorimetric assay section. (from page 3 line 108~121)

After the revision:

“Mecahnism of colorimetric assay”

“For colorimetric assay of glucose, the GOx precipitated in the detection zone which oxidizes glucose to gluconic acid while generating hydrogen peroxide. Another enzyme occupied in the detection zone, HRP, converted hydrogen peroxide into oxygen. Simultaneously, 4-AAP was reacted with MAOS. At the end of the HRP-mediated reaction, a blue product was formed.[1] For colorimetric of albumin, the TBPB precipitated in the detection zone which reacts like a pH indicator. The higher the amount of albumin, the more it reacts with TBPB, causing a color change from yellow to blue as the pH increases.[2] For colorimetric for Chol., the CO and CE precipitated in the detection zone which oxidizes Chol. to Chol-4-en-3-one while generating hydrogen peroxide. Another enzyme occupied in the detection zone, HRP, converted hydrogen peroxide into oxygen. Simultaneously, 4-AAP was reacted with ADOS. At the end of the HRP-mediated reaction, a pink product was formed. For colorimetric of ALP, the para-nitrophenylphosphate precipitated in the detection zone. As albumin and para-nitrophenylphosphate react, a yellow product, p-nitrophenylate, was formed. The remaining AST, ALT, creatinine, and UN reaction follows the commercial assay kit mechanism”

Round 2

Reviewer 1 Report

Authors have adressed all remarks.

Some small remarks:

Line 103: Mecahnism = Mechanism

Line 112: "For colorimetric..." detection? "of albumin, ..."

Author Response

Reviewer #1

Comment 1.

Line 103: Mecahnism = Mechanism

Response

Thank you. We correct them.

Reviewer 2 Report

The revision cover all the aspects.  Ok in the present form

Author Response

Reviewer #2

Comment 1.

The revision cover all the aspects.  Ok in the present form

Response

Thank you!
